# The Inhibitory Role of Rab11b in Osteoclastogenesis through Triggering Lysosome-Induced Degradation of c-Fms and RANK Surface Receptors

**DOI:** 10.3390/ijms21249352

**Published:** 2020-12-08

**Authors:** Manh Tien Tran, Yuka Okusha, Yunxia Feng, Masatoshi Morimatsu, Penggong Wei, Chiharu Sogawa, Takanori Eguchi, Tomoko Kadowaki, Eiko Sakai, Hirohiko Okamura, Keiji Naruse, Takayuki Tsukuba, Kuniaki Okamoto

**Affiliations:** 1Department of Dental Pharmacology, Graduate School of Medicine, Dentistry and Pharmaceutical Sciences, Okayama University, 2-5-1 Shikata-cho, Kita-ku, Okayama 700-8525, Japan; trantienmanh1508@gmail.com (M.T.T.); yokusha@bidmc.harvard.edu (Y.O.); yunxiafeng0302@163.com (Y.F.); 13898129859@163.com (P.W.); caoki@md.okayama-u.ac.jp (C.S.); eguchi.takanori@gmail.com (T.E.); 2Division of Molecular and Cellular Biology, Department of Radiation Oncology, Beth Israel Deaconess Medical Center, Harvard Medical School, Boston, MA 02115, USA; 3College of Basic Medicine, China Medical University, Shenyang 110122, China; 4Department of Cardiovascular Physiology, Graduate School of Medicine, Dentistry and Pharmaceutical Sciences, Okayama University, 2-5-1 Shikata-cho, Kita-ku, Okayama 700-8558, Japan; mmorimatsu@okayama-u.ac.jp (M.M.); knaruse@md.okayama-u.ac.jp (K.N.); 5Department of Endodontics, School of Stomatology, China Medical University, Shenyang 110002, China; 6Advanced Research Center for Oral and Craniofacial Sciences, Graduate School of Medicine, Dentistry and Pharmaceutical Sciences, Okayama University, 2-5-1 Shikata-cho, Kita-ku, Okayama 700-8525, Japan; 7Department of Frontier Oral Science, Graduate School of Biomedical Sciences, Nagasaki University, 1-7-1 Sakamoto, Nagasaki 852-8588, Japan; tomokok@nagasaki-u.ac.jp; 8Department of Dental Pharmacology, Graduate School of Biomedical Sciences, Nagasaki University, 1-7-1 Sakamoto, Nagasaki 852-8588, Japan; eiko-s@nagasaki-u.ac.jp (E.S.); tsuta@nagasaki-u.ac.jp (T.T.); 9Department of Oral Morphology, Graduate School of Medicine, Dentistry and Pharmaceutical Sciences, Okayama University, 2-5-1 Shikata-cho, Kita-ku, Okayama 700-8525, Japan; hiro-okamura@okayama-u.ac.jp

**Keywords:** Rab11b, c-Fms, RANK, NFATc-1, osteoclasts, vesicular transport

## Abstract

Rab11b, abundantly enriched in endocytic recycling compartments, is required for the establishment of the machinery of vesicle trafficking. Yet, no report has so far characterized the biological function of Rab11b in osteoclastogenesis. Using in vitro model of osteoclasts differentiated from murine macrophages like RAW-D cells or bone marrow-derived macrophages, we elucidated that Rab11b served as an inhibitory regulator of osteoclast differentiation sequentially via (i) abolishing surface abundance of RANK and c-Fms receptors; and (ii) attenuating nuclear factor of activated T-cells c1 (NFATc-1) upstream signaling cascades, following RANKL stimulation. Rab11b was localized in early and late endosomes, Golgi complex, and endoplasmic reticulum; moreover, its overexpression enlarged early and late endosomes. Upon inhibition of lysosomal function by a specific blocker, chloroquine (CLQ), we comprehensively clarified a novel function of lysosomes on mediating proteolytic degradation of c-Fms and RANK surface receptors, drastically ameliorated by Rab11b overexpression in RAW-D cell-derived osteoclasts. These findings highlight the key role of Rab11b as an inhibitor of osteoclastogenesis by directing the transport of c-Fms and RANK surface receptors to lysosomes for degradation via the axis of early endosomes-late endosomes-lysosomes, thereby contributing towards the systemic equilibrium of the bone resorption phase.

## 1. Introduction

Osteoclasts, multinucleated cells differentiated from macrophage-monocyte cells, play a critical role in bone tissue destruction [1,2]. Osteoclast differentiation, also called osteoclastogenesis, is directly induced via binding of the receptor activator of nuclear factor kappa-B ligand (RANKL) to the cell surface RANK receptor. Mechanistically, this binding stimulates six various signaling cascades essential for osteoclast differentiation including (i) the nuclear factor of activated T cells cytoplasmic-1 (NFATc-1); (ii) nuclear factor kappa B (NF-κB); (iii) phosphatidylinositol 3-kinase (PI3K/Akt); (iv) Jun N-terminal kinase (JNK); (v) extracellular signal-regulated kinase (Erk); and (vi) p38 mitogen-activated protein kinase (MAPK) [3], thereby boosting the secretion of bone-resorbing enzymes such as tartrate-resistant acid phosphatase (TRAP), Cathepsin K (CTSK), and matrix metalloproteinase 9 (MMP9), into the extracellular environment [4,5]. In addition, binding of monocyte/macrophage colony stimulating factor (M-CSF) to the cell surface c-Fms receptor is also essential for survival and differentiation of premature and mature osteoclasts [3,6]. The network of intracellular vesicle trafficking amongst subcellular compartments is evolutionarily conserved in eukaryotic cells [3,7].

Rab GTPases, specifically localized in integral membranes of intercellular organelles, are emerged as the central regulators of interrelated processes of membrane trafficking including budding, motility, docking, and fusion of vesicle, and are cardinal in transporting cargos to accurate destinations, cooperatively coupled to receptor signaling pathways [7,8]. Dynamic modification of Rab GTPases between inactive [Guanine Diphosphate (GDP)-bound] and active [Guanine Triphosphate (GTP)-bound] forms are catalyzed by specific enzymes. More particularly, guanine exchange factors (GEFs) catalyze the conversion of GDP-bound to GTP-bound form to switch on various cellular signals whereas GTPase activating proteins (GAPs) inactivate Rab GTPases via catalyzing GTP hydrolysis of GTP-bound form [9,10,11,12]. The Rab11 subfamily is structurally categorized into three closely related isoforms, Rab11a, Rab11b, and Rab25. Of these, Rab11a is copiously expressed; Rab11b is exclusively enriched in heart, brain, and testes [13], and in polarized MDCK, gastric parietal cells [14] whereas Rab25 is only found in the epithelial cells [15]. In rat osteoclasts, subcellular localization of Rab11b was identified to localize in perinuclear recycling compartments wherein functionally it was engaged in ruffled border membrane turnover and in osteoclast motility [14].

Our previous study clarified a vital role of lysosomal Rab44 as a decelerator of osteoclastogenesis via declining endogenous levels of c-Fms and RANK, followed by inactivation of NFATc-1 upstream signaling cascades through a poorly understood mechanism [16]. Moreover, our subsequent report disclosed the regulatory function of Rab27A on (i) directing transport routes of lysosome-related organelles to the ruffled border membrane, and (ii) weakening abundance of surface RANK and c-Fms receptors, thereby causing debilitation of osteoclast differentiation [17]. Recently we revealed a novel role of Rab11a in negatively regulating osteoclast differentiation mainly through down-regulating the surface abundance of c-fms and RANK receptors [18]. In this report, during the investigation of physiological roles of Rab11b for regulation of osteoclastogenesis, we also attempted to thoroughly elucidate a crucial role of lysosomes in the proteolytic degradation of c-Fms and RANK surface receptors in osteoclasts. Together, our findings stress the critical role of Rab11b for inhibition of osteoclastogenesis by directing the transport of c-Fms and RANK surface receptors to lysosomes for degradation via the axis of early endosomes-late endosomes-lysosomes; more importantly to provide a universal molecular mechanism by which Rab11b modulates osteoclastic bone resorption involved in the systemic homeostasis of boss mass and remodeling.

## 2. Results

### 2.1. Rab11b Is Up-Regulated at a Late Stage of Osteoclast Differentiation

To investigate the regulatory function of Rab11b during osteoclast differentiation, we initially screened the dynamic modifications of the mRNA and protein levels of Rab11b as well as several osteoclast markers such as c-Fos, NFATc-1, and CTSK in RAW-D cells and bone marrow-derived macrophages (BMMs) in the time courses of RANKL stimulation. Albeit the mRNA level of Rab11b was insignificantly altered (Figure 1A), the protein level thereof was strongly increased from day two to five (Figure 1C and Appendix A) in RAW-D cells upon five days of RANKL stimulation. Besides, levels of c-Fos and NFATc-1 were significantly strengthened from zero to two days, but drastically decreased from day three to five (Figure 1C and Appendix A). Similar effects on mRNA of Rab11b (Figure 1B), and protein levels of Rab11b, c-Fos, NFATc-1, and CTSK (Figure 1D and Appendix A) were observed in BMMs upon 4 days of RANKL stimulation. Importantly, our TRAP staining results showed that numbers of TRAP-positive multinucleated osteoclasts (MNCs) formed reached their maximum at days three and four in both cases of RAW-D cells (Figure 1E and Appendix A) and BMMs (Figure 1F and Appendix A), but somewhat reduced at day five in RAW-D cell-derived osteoclasts, suggesting that mature osteoclasts, unless otherwise specified, were formed in vitro after three days of RANKL stimulation. Altogether, these results indicated that Rab11b was strongly up-regulated at a late stage of osteoclast differentiation.

### 2.2. Rab11b Silencing Markedly Enhances Osteoclastogenesis

As above (Figure 1), Rab11b up-regulation was observed as mature osteoclasts were formed, it was surmised that Rab11b feasibly participated in inhibiting osteoclastogenesis. Thus, we first analyzed if osteoclastogenesis was affected by siRNA-mediated Rab11b suppression. Our data showed that Rab11b silencing promoted osteoclast formation in size (Figure 2A) and number (Figure 2B,C); more importantly, augmented the bone-resorbing activity (Figure 2D,E) in osteoclasts differentiated from RAW-D cells. The same effects were also found in BMM-derived osteoclasts (Appendix A). In addition, Rab11b silencing enhanced expression levels of c-Fms, RANK, NFATc-1, and CTSK (Figure 2F) in osteoclasts differentiated from RAW-D cells (left panel) and BMMs (right panel). From these observations, we conjectured that Rab11b was a negative regulator of osteoclastogenesis.

Because Rab11a and b share ~90% amino acid homology [13], it was crucial to confirm if Rab11b-targeting siRNA altered mRNA and protein levels of Rab11a. Intriguingly, our results showed that while Rab11b silencing slightly reduced, but insignificantly, mRNA levels of Rab11a (Appendix A), it still increased expression levels of Rab11a (Appendix A) in osteoclasts derived from both RAW-D cells and BMMs. These findings did clarify that Rab11b silencing rescued the expression level of Rab11a, suggesting that both homologs appeared to be functionally interrelated for the regulation of osteoclast differentiation.

### 2.3. Rab11b Overexpression Significantly Abolishes Osteoclastogenesis

In order to further elucidate the regressive role of Rab11b in osteoclastogenesis, we next evaluated if Rab11b overexpression altered osteoclast differentiation. We generated RAW-D cells stably expressing GFP, referred to as the control, and GFP-tagged Rab11b (GFP-Rab11b) with three different clones, abbreviated by #1, #2, and #3, referred to as Rab11b overexpression. As we expectated, the bands showing GFP (Figure 2L, 6th panel from the top, 1st lane) and GFP-Rab11b (Figure 2L, 5th and 6th panel from the top, indicated by black arrows) expressions were detected in osteoclasts derived from RAW-D cells. Next, assessing osteoclast formation by TRAP staining, we found that Rab11b overexpression (i) diminished osteoclast size (Figure 2G), (ii) decreased the number of the TRAP-positive MNCs (Figure 2H), especially that of MNCs containing more than 10 nuclei (Figure 2I), and (iii) abolished the bone-resorbing ability of osteoclasts (Figure 2J,K) derived from RAW-D cells. In addition, expression levels of osteoclast markers, including c-Fms, RANK, and CTSK were markedly decreased whereas that of NFATc-1 was increased with respect to Rab11b overexpression (Figure 2L). Cumulatively, these results further postulated a critical role of Rab11b in negatively regulating osteoclastogenesis. By virtue of high structural and sequential homologies between Rab11a and b [13]; therefore, in order to ensure the authenticity of our results obtained so far, we tested if the primary antibody used to detect Rab11b was able to cross-react to Rab11a. As expected, our results showed that there was no cross-reaction of the primary antibody of Rab11b to Rab11a (Appendix A, the top panel, red rectangle). Moreover, we observed that Rab11b overexpression declined expression level of Rab11a (Appendix A, the top panel, purple rectangle), and reversely (Appendix A, middle panel, yellow rectangles), further illuminating their interdependently functional roles in osteoclasts.

### 2.4. Rab11b Overexpression Triggers Ca^2+^-Dependent NFATc-1 Stabilization

Induction and activation of NFATc-1 [19,20] promoted *CTSK* gene transcription [21] during osteoclast differentiation. Notably, Rab11b overexpression weakened endogenous levels of c-Fms, RANK, and CTSK, but strengthened that of NFATc-1 in osteoclasts (Figure 2L). To unveil such an extraordinary phenomenon, we first assessed the dynamic modifications of NFATc-1 and CTSK over a time course of RANKL stimulation in RAW-D cells expressing GFP or GFP-Rab11b. Indeed, our data showed that Rab11b overexpression (i) decreased endogenous CTSK, but (ii) drastically enhanced that of NFATc-1 on day three (Figure 3A), relevant to our initial findings (Figure 2L). NFATc-1, a master regulator of osteoclast differentiation [22], could translocate into nuclei to transcriptionally promote osteoclast specific genes [19,22]. By nuclear/cytoplasmic fractionation, we found Rab11b overexpression enhanced cytosolic level, but decreased the nuclear level of NFATc-1 (Figure 3B). Similarly, to ensure our data consistency, we also investigated the effects of Rab11b silencing on regulating NFATc-1 and CTSK in osteoclasts derived from RAW-D cells. Indeed, Rab11b silencing markedly increased expression levels of these proteins in a time course of RANKL stimulation (Appendix A), and accelerated nuclear translocation of NFATc-1 (Appendix A), suggesting one potentiality of Rab11b overexpression-mediated NFATc-1 regulation independent of osteoclastogenesis signaling pathways induced by RANKL.

Previous reports revealed that NFATc-1 deubiquitination was mediated by transient elevation of intracellular Ca^2+^ ((Ca^2+^_i_)) [22,23,24]. We thus examined if cytosolic NFATc-1 accumulation by Rab11b overexpression initially resulted from the transient elevation of (Ca^2+^_i_) in osteoclasts. Our data showed that Rab11b overexpression markedly enhanced (Ca^2+^_i_) levels in RAW-D cells stimulated with RANKL for two days (Figure 3C,D). Furthermore, we tested transient (Ca^2+^_i_) oscillation by ionomycin, a selective Ca^2+^, efficacious in Ca^2+^ fluxes [25]. Our data showed that ionomycin treatment drastically elevated (Ca^2+^_i_) with respect to Rab11b overexpression (Appendix A). More critically, Rab11b overexpression decreased mRNA levels of NFATc-1 (Figure 3E), but the increased endogenous level of NFATc-1, and abolished NFATc-1 polyubiquitination at lysine 48 (Figure 3F), suggesting a novel mechanism of Rab11b overexpression-mediated regulation of (Ca^2+^_i_)-dependent NFATc-1 stabilization, following RANKL stimulation.

### 2.5. Rab11b Modulates NFATc-1 Signaling Cascades in the RAW-D Cells and BMMs upon Stimulation with RANKL and M-CSF, Respectively

Due to Rab11b-mediated declination of endogenous levels of c-Fms and RANK (Figure 2L), we, therefore, tested how NFATc-1 upstream signaling cascades would be altered with respect to Rab11b overexpression. Indeed, our results indicated that Rab11b overexpression caused the phosphorylation levels of p38, IκBα, Akt, and Erk to be strongly decreased while that of JNK was indistinguishable (Figure 4A). Furthermore, we earlier showed that Rab11b silencing activated NFATc-1 (Figure 2F and Appendix A), we, therefore, examined the effects of Rab11b silencing on the osteoclastogenic signaling cascades in BMMs over a time course of M-CSF stimulation. Indeed, our results showed that phosphorylation levels of p38, Akt, and Erk were significantly increased in Rab11b-silenced BMMs (Figure 4B). Together, these data elucidated the stimulatory effects of Rab11b silencing on modulating osteoclastogenic signaling cascades, thereby promoting osteoclast differentiation.

### 2.6. GFP-Rab11b Localized in Early and Late Endosomes, Golgi Complex, and Endoplasmic Reticulum Causes the Enlargement of Early and Late Endosomes in RAW-D Cells and Osteoclasts

An earlier report identified Rab11b required for the transport of internalized transferrin to the plasma membrane in CHO cells localizes in endocytic recycling compartments (ERCs) [26]; however, little is known about the subcellular localization of Rab11b in RAW-D cells and osteoclasts. To address it, co-localization of the stably expressed GFP-Rab11b was examined with several organelle-specific marker proteins such as Rab5 (early endosomes, EEs), Rab7 (late endosomes, LEs), GM130 (Golgi complex), LAMP1 (lysosomes), and KDEL (endoplasmic reticulum, ER). Our results revealed that Rab11b was co-localized with Rab5, Rab7, GM130, and KDEL, but not LAMP1 in both RAW-D cells (Appendix A), and RAW-D-derived osteoclasts (Figure 5A–C). Of note, we observed that Rab11b overexpression caused stronger intensity of (i) Rab5-positive fluorescence and enlarged Rab5-positive EEs in RAW-D cells (Appendix A) and osteoclasts (Figure 5D), and (ii) those of Rab7-positive LEs in RAW-D cells (Appendix A) and osteoclasts (Figure 5E), but not (iii) those of GM130-positive Golgi complex in RAW-D cells (Appendix A), and in osteoclasts (Figure 5F). From these observations, we conjectured that size-based enlargement of EEs and LEs possibly resulted from the accumulation of internalized cargos in EEs and LEs in both RAW-D cells and osteoclasts differentiated from RAW-D cells, suggesting that, regardless of RANKL stimulation, Rab11b overexpression was sufficient for such a morphological alteration in EEs and LEs.

### 2.7. Rab11b-Mediated Augmentation of Lysosomal Function Regulates Endogenous Turnovers of c-Fms and RANK in Osteoclasts

Previous studies comprising ours have highlighted the functional importance of lysosomes in osteoclast differentiation [17,27,28,29]; hence, we tested if Rab11b alteration could affect the function of the lysosomal system in osteoclasts. Experimentally, our results showed Rab11b overexpression resulted in markedly elevated expression levels of lysosomal LAMP1, and Cathepsins B and D (Figure 6A) whereas its suppression insignificantly altered patterns of such lysosomal markers (Appendix A), prompting that Rab11b was engaged in regulating lysosomal functions in osteoclasts. Because Rab11b overexpression decreased expression levels of c-Fms and RANK, we examined if lysosomes were involved in c-Fms and RANK degradation in osteoclasts. To address this question, we analyzed the effects of lysosomal inhibition by a lysosomal blocker, chloroquine (CLQ), in osteoclasts. Our results showed lysosomal inhibition increased endogenous levels of c-Fms and RANK, but not Rab11b, in dose- and time-dependent manners (Figure 6B,C); more importantly, no significant alteration of mRNA levels of c-Fms (Figure 6D) and RANK (Figure 6E) was observed between each group treated with or without CLQ over a time course in osteoclasts expressing GFP or GFP-Rab11b. In addition, we found that Rab11b overexpression substantially declined levels of c-Fms (Figure 6D) and RANK (Figure 6E) mRNAs. Since proteasome complex is thought to be responsible for cellular proteolysis [30,31], we used MG132, a peptide aldehyde proteasome inhibitor, to analyze the effects of proteasome inhibition on endogenous levels of c-Fms and RANK. Amazingly, no marked alternation of patterns of c-Fms and RANK were found in response to MG132 treatment (ranging from 0 to 20 µM) (Appendix A). These data indicated that regulation of c-Fms and RANK by lysosomal inhibition, but not by proteasome inhibition, was likely due to their post-translational regulation in osteoclasts.

To further explore the detailed mechanism of lysosome-mediated proteolysis of c-Fms and RANK, it was essential to inhibit new protein synthesis by cycloheximide (CHX). We initially determined an optimal concentration of CHX used for the subsequent experiments. Our results showed an inhibitory effect of CHX (20 µg/mL) on endogenous levels of RANK and c-Fms was stabilized after 3 h treatment (Appendix A) in osteoclasts expressing GFP and GFP-Rab11b. Treatment of (CHX + CLQ) combination markedly increased endogenous levels of c-Fms and RANK as compared to those treated with CHX alone (Figure 6F), regardless of Rab11b overexpression. Moreover, no significant alteration of patterns of c-Fms and RANK were found in each group treated with CHX-alone or with (CHX + MG132) combination (Appendix A). In addition, no cytotoxic effects were found in all cases treated with the drug(s) (Appendix A). Taken together, these data emphasized an important role of lysosomes in regressing osteoclast differentiation through mediating proteolysis of c-Fms and RANK.

### 2.8. Rab11b Overexpression Promoted Lysosome-Mediated Degradation of c-Fms and RANK Receptors in Osteoclasts

As above, Rab11b localization was observed in early and late endosomes. One of the well-characterized transport routes of lysosome-degraded cargos, including surface receptors, is sequentially coated, sorted, and processed by early and late endosomes [32,33,34]. Consequently, we speculated that Rab11b promoted the delivery of c-Fms and RANK receptors from early, late endosomes to lysosomes in osteoclasts. We first evaluated how Rab11b alteration affected surface levels of c-Fms and RANK receptors in osteoclasts. Our results showed Rab11b overexpression strongly decreased surface levels of c-Fms and RANK receptors (Figure 7A,B) whereas its silencing rescued their surface levels (Figure 7C,D), unveiling a suppressive effect of Rab11b on surface levels of c-Fms and RANK receptors in osteoclasts. Next, we analyzed if Rab11b promoted lysosome-mediated degradation of c-Fms and RANK receptors. By pre-treating osteoclasts with a combination of CHX (20 µg/mL) and MG132 (10 µM), we found that endogenous levels of c-Fms and RANK were rapidly decreased in response to Rab11b overexpression (Figure 7E–G), and no cytotoxic effects on osteoclast viability were observed in all treated cases (Figure 7H). Altogether, these results illuminated a novel function of Rab11b as an accelerator of the lysosome-mediated degradation of c-Fms and RANK receptors in osteoclasts.

## 3. Discussion

In this study, we have provided compelling results elucidating that Rab11b was up-regulated at a late stage of osteoclast differentiation and maturation. Notably, Rab11b mRNA level (Figure 1A,B) was insignificantly varied during osteoclast differentiation, suggesting it was highly possible that Rab11b could be stabilized by post-translational modification, for instance, by deubiquitinases. Nonetheless, searching for potential candidates serving Rab11b deubiquitination has been under investigation. Here, by loss and gain of Rab11b expression experiments, Rab11b, for our first time, has been identified as a negative regulator of osteoclast differentiation. Indeed, Rab11b silenced an increased number of multinucleated and giant osteoclasts, enhanced bone resorbing activity, and strengthened endogenous levels of osteoclast markers (Figure 2 and Appendix A) while Rab11b overexpression inhibited such properties of osteoclasts (Figure 2). Intriguingly, we found that Rab11b overexpression triggered Ca^2+^-dependent NFATc-1 deubiquitination and stabilization (Figure 3).

Previous reports have disclosed the physiological role of transient elevation of intracellular Ca^2+^ in sequential events of dephosphorylation, deubiquitination, stabilization, and nucleus translocation of NFATc-1 and that it was crucial for promoting transcriptional processes of osteoclastogenic genes such as CTSK, TRAP, and MMP9 [21,35,36]. Yet, it was surprising that, by Rab11b overexpression, the cytoplasmic NFATc-1 level was increased while nuclear NFATc-1 level was decreased in osteoclasts (Figure 3B). These data raised questions of how Rab11b overexpression could accumulate NFATc-1 in the cytosol, and what the potent mechanism of nuclear NFATc-1 translocation regulation was. Importantly, Rab11b overexpression declined surface abundance of c-Fms and RANK receptors (Figure 7A), thereby attenuating NFATc-1 upstream signaling cascades (Figure 4), raising one feasibility that nuclear NFATc-1 translocation could be controlled by unknown factors that were regulated by certain NFATc-1 upstream signaling pathways dependent on c-Fms/RANK receptor activation. Resultantly, our future works are to identify such important factors so as to further understand underlying molecular mechanisms of NFATc-1 regulation during osteoclast differentiation.

Rab11b was mainly localized in early and late endosomes in RAW-D cells (Appendix A) and osteoclasts (Figure 5A); furthermore, Rab11b overexpression caused size-based enlargement of early and late endosomes, consistent with our initial findings, in which Rab11b was up-regulated at a late stage of osteoclast differentiation. Furthermore, it was possible that Rab11b up-regulation accelerated cargo internalization into early and late endosomes. Surprisingly, an earlier study revealed CRISPR/Cas9-based depletion of either Rab11a or b caused enlargement of early endosomes, and enhancement of late endosomal and lysosomal activities in non-polarized HeLa cells [37], contradictory to our observations in osteoclasts, indicating a broad spectrum of Rab11b functional features in human cells. Although it was clarified that Rab11b did not localize in lysosomes (Figure 5), Rab11b overexpression up-regulated LAMP1, and Cathepsins B and D in osteoclasts (Figure 6A), proposing that Rab11b was possibly linked to lysosomal functions. Besides, we also observed Rab11b localization in Golgi complex, and ER in both RAW-D cells and osteoclasts; nonetheless, how it was functionally linked to cargo delivery amongst these organelles in osteoclasts has been completely obscure. One of our speculations is that Rab11b is probably engaged in regulating other osteoclast receptors, required for osteoclastogenesis regulation, by the direct- or indirect-transport of intracellular receptors to cell surfaces, distinct from the transport route of c-Fms and RANK surface receptors via the axis of early endosomes-late endosomes-lysosomes. Calcitonin receptor (CTR), for instance, directly binds to calcitonin, a calcium-lowering hypocalcemic hormone secreted by the thyroidal C [38,39], was demonstrated to trigger osteoclast retraction, abnormally transfigured osteoclast morphology [40], and finally inhibition of bone resorption [41,42]. Astonishingly, CTR expression was enhanced before the formation of RANKL-induced multinucleated osteoclasts [43,44]. Therefore, one of our future research goals would be examining if Rab11 is functionally involved in regulation of surface abundance of CTR or CTR-like receptor (CLR) during osteoclast differentiation.

Albeit that lysosomal involvement in osteoclast differentiation regulation has been highlighted [17,28,29], a lysosomal role for proteolytic cleavage of c-Fms and RANK has not been previously reported. In our study, using a specific lysosomal inhibitor, we revealed a crucial role of lysosomes in c-Fms and RANK proteolysis, regardless of Rab11b overexpression (Figure 5B,C,F), suggesting a natural characteristic of lysosomes as a suppressive regulator of osteoclast differentiation. To prove lysosomal vitality to proteolysis of c-Fms and RANK, we also evaluated the effect of MG132-mediated inhibition of proteasome complex upon proteolytic cleavage of c-Fms and RANK. Different from our observations in lysosomes, proteasome complex inhibition did not alter endogenous levels of c-Fms and RANK, suggesting that neither c-Fms nor RANK could undergo post-translational modifications by ubiquitin-proteasome complex. These data further strengthened our observational theory of a unique lysosomal function on proteolysis of c-Fms and RANK in osteoclasts. It is worth noting that reduction of the surface abundance of c-Fms and RANK receptors results in weakening osteoclastogenic signal transduction, thereby debilitating osteoclastogenesis [6,45,46,47,48]. Interestingly, our data revealed Rab11b overexpression decreased surface abundance of c-Fms and RANK receptors (Figure 7A,B) via ameliorating proteolysis of c-Fms and RANK (Figure 7E), further substantiating a suppressive role of Rab11b in osteoclast differentiation.

Because the individual Rab11 family members have not been broadly studied, it has been unclear that Rab11a and b have overlapping or distinct functions in osteoclasts. Whether Rab25, found to regulate integrin expression in colonic epithelial cells [15], is present in osteoclasts has been controversial. Interestingly, our recent study elucidated that Rab11a played an important role in negatively regulating osteoclast differentiation [18]; we therefore attempted to investigate the functional conjunction between Rab11a and b, using loss and gain of Rab11b expression experiments. As predicted, siRNA-mediated Rab11b suppression rescued the Rab11a expression in osteoclasts derived from RAW-D cells (Appendix A) and BMMs (Appendix A). Furthermore, Rab11b overexpression weakened the expression level of Rab11a, and reversely (Appendix A), suggests that both homologs (i) are functionally interdependent of each other and (ii) are negative regulators of osteoclast differentiation. In addition, we also hypothesized that other Rab GTPases might have functionally interconnected and cooperated with each other for the regulation of osteoclastogenesis. To Rab11b, it might be either Rab11a, unknown factors, or both (Figure 8).

In conclusion, our study, the first report to shed light on the critical role of Rab11b as a negative regulator of osteoclast differentiation via directing and transporting c-Fms and RANK receptors via a selective axis of early endosomes-late endosomes-lysosomes for proteolytic degradation, subsequently abolishing osteoclastogenesis, eventually stabilizing the osteoclastic bone resorption phase. The study of how inhibitory regulators worked on osteoclast differentiation was beyond our knowledge of macrophagic features and osteoclastic bone resorption. Finally, our novel findings may contribute to facilitate the establishment of either efficacious drugs, potential therapeutic methods, or both, for the treatment of macrophage-caused inflammatory diseases related to bone loss, consisting of osteoporosis, Paget’s disease, periodontitis, and rheumatoid arthritis, all of which are characterized by excessive bone resorption by gain of function in osteoclasts [49,50,51,52]. Over the past decade, the introduction of target biologic therapy having significantly improved the clinical outcomes for patients with bone-related diseases abovementioned, is mediated by RANKL blockade [50] or by targeting pro-inflammatory cytokines [53] in order to prevent osteoclast differentiation, which subsequently reduces bone erosion. However, our study has comprehensibly elucidated a novel insight of how the bone-resorbing capacity of osteoclasts was inhibited by a mechanism underlying Rab11b-mediated lysosomal degradation of c-Fms and RANK surface receptors, raising one probability of the drug development program for treatment of the bone-related diseases caused by gain of function in osteoclasts aimed at either activating lysosomal function in combination with RANKL blockade in pre-osteoclasts or upregulating Rab11 in both pre-osteoclasts and osteoclasts.

## 4. Materials and Methods

### 4.1. Antibodies and Reagents

Isolation and purification of recombinant RANKL was prepared by the protocols described heretofore [54]. M-CSF was purchased from Kyowa Kogyo (Tokyo, Japan). Rabbit polyclonal anti-Cathepsins B and D, K antibodies were purified as the protocol described previously [55]. The following antibodies were used in this study: rat monoclonal anti-LAMP1 (Cat. No. 553792, BD Biosciences, NJ, USA), rabbit polyclonal anti-c-Fms (Cat. No. sc-692, Santa Cruz, CA, USA), mouse monoclonal anti-c-Fms (#G1019) (Santa Cruz Biotechnology, Santa Cruz), mouse monoclonal anti-c-Fos (#1019) (Santa Cruz Biotechnology, Santa Cruz, CA 95060, USA), mouse monoclonal anti-NFATc-1 (Cat. No. sc-7294, Santa Cruz, CA, USA), mouse monoclonal anti-RANK (NBP2-247-2, Novus Biologicals Europe, Abingdon, UK), rabbit polyclonal anti-GFP (Green Fluorescent Protein) (Medical & Biological Laboratories Co., LTD., Nagoya, Japan), rabbit polyclonal anti-Rab11a (#2413), and rabbit polyclonal anti-Rab11b (#2414), rabbit monoclonal anti-Rab5 (#3547), rabbit monoclonal anti-Rab7 (#9367) (Cell Signaling, Danvers, MA, USA), mouse monoclonal GM130 (Cat. no. 610823) (BD, Biosciences, Franklin Lakes, NJ, USA), mouse monoclonal anti-KDEL (Cat. no. ADI-SPA-827) (Enzo Life Sciences, Farmingdale, NY, USA), rabbit monoclonal anti-phospho-p38 (Cat. no. 4511S, Thr180/Tyr182), rabbit polyclonal anti-p38 MAPK (Cat. no. 9212S), rabbit monoclonal anti-phospho-IκBα (Cat. no. 2859S, Ser32), rabbit monoclonal anti-IκBα (Cat. no. 4812S), rabbit polyclonal anti-phospho-JNK (Cat. no. 9251S, Thr183/Tyr185), rabbit polyclonal anti-JNK (Cat. no. 9252S), rabbit monoclonal anti-phospho-Akt (protein kinase B) (Cat. no. 4060S, Ser473), rabbit monoclonal anti-Akt (Cat. no. 4691S), rabbit polyclonal anti-phospho-Erk1/2 (Cat. no. 9101S, Thr202/Tyr204), rabbit polyclonal anti-Erk1/2(Cat. no. 9102S), mouse monoclonal anti-ubiquitin (#8017) Santa Cruz Biotechnology, Santa Cruz, CA 95060), rabbit polyclonal K48-linked ubiquitin (#4289S) (Cell, Signaling, Danvers, MA, USA), rabbit monoclonal anti-GAPDH (Cat. no. 2118S), Alexa Fluor 488 goat anti-rabbit IgG, Alexa Fluor 488 goat anti-mouse IgG, and Alexa Fluor 594 goat anti-rat IgG (Cell Signaling Technology Danvers, MA, USA). DAPI (4′,6-diamidino-2-phenylindole) and other reagents purchased are as follow: Cal-590TM AM (COSMO bio Co., Ltd., Japan) and Ionomycin (WAKO, Osaka, Japan). Other reagents were purchased from Sigma-Aldrich (Tokyo, Japan). The Osteo Assay Stripwell Plate was purchased from Corning, Inc. (Corning, NY, USA).

### 4.2. Cell Culture

A murine monocytic cell line (also known as RAW-D) gifted from Toshio Kukita (Kyushu University, Japan) [56,57] was cultured in minimum essential medium α (MEMα) (Wako Pure Chemicals, Osaka, Japan) supplemented with 10% fetal bovine serum (FBS), penicillin (100 U/mL) and streptomycin (100 mg/mL). For osteoclast differentiation from RAW-D cells, RAW-D cells were cultured in MEMα with RANKL (300 ng/mL) for 3 days. Bone marrow-derived macrophages (BMMs) were isolated from the femurs and tibias of 5-week-old male C57BL/6J mice (SLC, Shimizu Laboratory, Japan) as described previously [54] through flushing the bone marrow cavity, and subsequently cultured in MEMα with M-CSF (50 ng/mL) at 37 °C in 5% CO_2_. One day later, the non-adherent cells were collected and cultured in new MEMα with M-CSF (30 ng/mL). On day 3 of culturing, the adherent cells, referred to as BMMs, were refreshed with new MEMα with M-CSF (30 ng/mL). For osteoclast differentiation from BMMs, BMMs were cultured in MEMα with both M-CSF (30 ng/mL) and RANKL (300 ng/mL) for 3 days.

### 4.3. Quantitative Real-Time Polymerase Chain Reaction (RT-PCR) Analysis

For real-time PCR analysis, Trizol (Molecular Research Center, Cincinnati, OH, USA) was used for RNA extraction and purification from cultured cells. Following which, 0.1 μg of total RNA was reverse transcribed for cDNA synthesis using an iScript cDNA Synthesis Kit (Bio-Rad, Hercules, CA, USA). Quantitative real-time PCR was done using MJ Mini (BioRad), according to the manufacturer’s instructions. The primer sets used are indicated in Table 1.

### 4.4. Immunoblot Analysis

Cells were seeded and grown to confluence in either 60 mm dishes or 10 cm dishes, followed by appropriate treatments detailed in the result section. Cells were lysed by RIPA buffer (50 mM Tris-HCl (pH 8.0), 1% Nonidet P-40, 0.5% sodium deoxycholate, 0.1% SDS, 150 mM NaCl) supplemented with proteinase inhibitor cocktail (Sigma-Aldrich, Tokyo, Japan), put on ice for 30 min, and subsequently centrifuged for 20 min at 15,000 rpm. The protein concentrations were subsequently determined by Bio-Rad assay (Thermo Pierce, Rockford, IL, USA) according to the manufacturer’s guidance. The cell lysates ranging from 15 to 50 µg were run on 10% SDS-PAGE electrophoresis gels, and then transferred to polyvinylidene fluoride (PVDF) membranes by a wet-transfer method. The blots were blocked in Tris-buffered saline containing 0.05% Tween 20 and 5% skim milk for 1–1.5 h at room temperature (RT), and subsequently probed with various antibodies (1/1000) at 4 °C overnight. After being washed, the blots were incubated with appropriate horseradish peroxidase (HRP)-conjugated secondary antibodies (GE Healthcare). Blots were eventually reacted with ECL substrate (Millipore, Burlington, MA, USA). The immunoreactive bands were observed with a ChemiDoc MP Imaging System (Bio-Rad, Hercules, CA, USA). To confirm equal amounts of loaded samples, the blots were incubated with either an anti-GAPDH-HRP antibody for 1 h or an anti-Actin antibody (4 °C, overnight), and subsequently incubated with rabbit HRP-conjugated secondary antibody.

### 4.5. RNA Interference

RAW-D cells or BMMs cultured in 60 mm dishes (2 × 10^5^ cells/dish) for protein analysis or (1 × 10^5^ cells/dish) for RNA analysis, were transfected with 10 pmol of one of two independent duplex siRNAs (Rab11b) covering the targeted sequences: 5′-GGGACGACGAGUACGAUUACCUAUU-3′ (Rab11b siRNA #1), or 5′-AAUAGGUAAUCGUACUCGUCGUCCC-3′ (Rab11b siRNA #2) (Invitrogen Custom Primers, Invitrogen, Carlsbad, CA, USA), or with 10 pmol of non-targeting siControl (siCtrl) (Stealth RNAi siRNA Negative Control, Invitrogen, Carlsbad, CA, USA) diluted in Opti-MEM I (Life Technologies) with Lipofectamine RNAiMAX transfection reagent (Invitrogen, Carlsbad, CA, USA) at 37 °C overnight, according to the manufacturer’s instructions. At the 1st day of post-transfection, the cells were replaced and cultured with new culture media containing RANKL (300 ng/mL) for 3 days. The knockdown efficacy was assessed by qPCR, immunoblotting, and TRAP staining where appropriate. Only for bone resorption analysis, after siRNA transfection, the cells were cultured with culture media supplemented with RANKL (500 ng/mL) for 7–10 days.

### 4.6. TRAP Staining

Cells were fixed with 4% paraformaldehyde (PFA) at RT for 1 h, and subsequently treated with 0.2% Triton X-100 in PBS at RT for 5 min. Ultimately, cells were stained for TRAP solution including 0.01% naphthol AS-MX phosphate (Sigma-Aldrich, Tokyo, Japan) and 0.06% fast red violet LB salt (Sigma-Aldrich, Tokyo, Japan) in the presence of 50 mM sodium tartrate and 50 mM sodium acetate (pH 5.0). TRAP-positive multinuclear cells were counted under a light microscope, and the images were photographed by the Olympus FSX100 microscope. The multinuclear cells harboring three or more nuclei were considered as mature osteoclasts.

### 4.7. Immunocytochemistry

The cells seeded and grown on glass coverslips were fixed with 4.0% PFA in PBS for 1 h at RT. After washing with PBS, the fixed cells were permeabilized with 0.1% Triton X-100 in PBS for 10 min. The cells were incubated sequentially with 10% normal goat serum for 30 min and with primary antibodies at 4 °C overnight. The cells were washed, and stained with secondary antibodies, Alexa Fluor 594 goat anti-rat IgG or Alexa Fluor 594 goat anti-rabbit IgG (Cell Signaling Technology, Danvers, MA, USA). Ultimately, nuclear staining with DAPI (Invitrogen Carlsbad, CA, USA) was carried out. The samples were visualized using a laser-scanning confocal imaging system (LSM 780 META; Carl Zeiss, AG, Jena, Germany).

### 4.8. Bone Resorption Assay

The bone resorption assay was done by the method described previously. The bone-resorbing activity of osteoclasts derived from RAW-D cells and BMMs upon M-CSF (30 ng/mL) and RANKL (500 ng/mL) stimulation for 7–10 days was determined using the Osteo Assay Stripwell Plates. The images of the bone resorption area were photographed by Floid cell imaging station (Thermo Fisher, Waltham, MA, USA) and analyzed by Image J software.

### 4.9. Retrovirus Construction and Expression of Mouse Rab11b in RAW-D Cells

Retrovirus construction and expression of Rab11b were carried out as the methods described heretofore [16]. Briefly, the full-length cDNAs of mouse Rab11b were generated by PCR method employing cDNA originated from bone marrow macrophages (BMMs) simultaneously stimulated with M-CSF and RANKL for 72 h. The primers were used for GFP (for in-fusion) forward: 5′-GGACGAGCTGTACAAGGGCACCCGCGACGACGAGTAC-3′ and reverse: 5′-CTACCCGGTAGAATTCTTAGATGTTCTGACAGCACTGC-3′. Then, the cDNA(s) were amplified using Prime STAR GXL DNA polymerase (Takara, Tokyo) with 40 cycles at which denaturation at 94 °C for 10 s, annealing at 62 °C for 30 s, and extension at 72 °C for 3 min for each cycle. To generate GFP-Rab11b fusion protein, the amplified fragments were fused with a linearized pMSCVpuro-GFP, gifted by Kosei Ito (Nagasaki University, Japan), using In-Fusion cloning kit (Clontech, Mountain View, CA, USA). pMSCVpuro-GFP was also used as a control vector. GFP or GFP-Rab11b vectors were transfected into HEK293T cells by using Lipofectamine 2000 (Life Technologies, Gaithersburg, MD, USA), according to the manufacturer’s instructions. After incubation at 37 °C in 5% CO_2_ for 48 h, the supernatants composed of viruses were collected and subsequently infected into RAW-D cells. The cells were cloned by puromycin (5 μg/mL) diluted in MEMα supplemented with 10% FBS, penicillin (100 U/mL), and streptomycin (100 mg/mL), and the medium was refreshed every 3 days. After 2 weeks of culturing, puromycin-resistant cells were obtained and referred to as RAW-D cells expressing GFP or GFP-Rab11b.

### 4.10. CellTiter-Glo Viability Assay (CTG)

Cytotoxicity evaluation was carried out using the CellTiter-Glo Luminescent Cell Viability Assay Kit (Promega, Madison, WI, USA), according to the manufacturer’s protocol. Furthermore, 5 × 10^3^ cells/well were seeded and grown in 95 flat-bottomed well plates. The plate was incubated at 37 °C in 5% CO_2_ for 24 h prior to RANKL (300 ng/mL) supplementation. The plates were incubated for 3 days in 5% CO_2_, then simultaneously added with cycloheximide (CHX) (20 µg/mL) with or without chloroquine CLQ (10 µM), or with or without MG132 (20 µM) to each well, and subsequently incubated for the designated periods before being quenched with CellTiter-Glo (Promega, Madison, WI, USA, 50 µL/well), then centrifuged at 1000 rpm for 1 min and incubated at RT for 15 min. Luminescence was measured with a plate reader (Molecular Devices, San Jose, CA, USA).

### 4.11. Nuclear/Cytoplasmic Fractionation

The subcellular fractionation was carried out using NE-PER nuclear and cytoplasmic extraction kit (Thermo Scientific). Cells were harvested by using cell scrapers, lysed with CER I containing protease inhibitors, and then incubated on ice for 15 min. Permeabilized cells were subsequently added with ice-cold CER II, vortexed for 10 s, incubated on ice for 1 min, and centrifuged at 500× *g* for 10 min at 4 °C. The pellets were put on ice, and the supernatants were further centrifuged at 16,000× *g* for 20 min at 4 °C. Following which, 16,000× *g* supernatants were collected and referred to as the cytoplasmic fraction. After being washed twice with ice-cold PBS, the pellets were suspended in ice-cold NER containing protease inhibitors, and subsequently vortexed on the highest speed for 1 h at 4 °C. After centrifugation at 16,000× *g* for 15 min at 4 °C, the supernatants were collected and referred to as the nuclear fraction.

### 4.12. In Vitro Ubiquitination Assay

RAW-D cells (1 × 10^6^ cells) were seeded and grown in 10 cm culture dishes, then stimulated with RANKL (300 ng/mL). After 3 days of culture, the cells were washed twice by cold PBS, scraped by a scraper. Furthermore, 0.8–1.0 mg of whole cell lysates prepared by lysis in 150 mM NaCl, 1% Nonidet P-40, 1% deoxycholate, 0.1% SDS, 50 mM Tris (pH 7.5), 1 mM PMSF, 25 mM NaF, and protease inhibitors, was added with mouse-IgG or NFATc1 antibody on the rotator at 4 °C for 1–2 h. The immune complexes were incubated with protein-G beads overnight, preblocked with 10% bovine serum albumin. The immunoprecipitates were washed thrice with the same lysis buffer, and twice with IP buffer. The immunoprecipitates were mixed with 6× Laemmli dye, boiled at 60 °C for 20 min, and loaded on SDS-PAGE gels. After transfer, the blots were blocked with the 5% skim milk dissolved in TBS-T for 1–2 h, and probed with anti-ubiquitin antibody or anti-K48-ubiquitin antibody at 4 °C overnight. The blots were washed thrice with TBS-T, probed with secondary antibody sequentially, and developed.

### 4.13. Surface Biotinylation Assay

RAW-D cells (5 × 10^5^ cells) were seeded and grown in a 10 cm dish for 3 days upon RANKL (300 ng/mL) stimulation. Cells were washed twice with cold PBS, and subsequently incubated for 1 h at 4 °C with 3.0 mg/mL Sulfo-NHS-SS-Biotin (Pierce) in DPBS+. Cell dishes were gently rinsed in 100 mM glycine (10 min, 3×), and subsequently in 20 mM glycine (10 min, 3×), both in DPBS+. Cells were harvested, and lysed by the buffer LB3 encompassing 50 mM Tris/HCl (pH 7.4), 150 mM NaCl, 1 mM EDTA, 1% (w/v) Triton X-100, and protease inhibitor. The cell lysates were gently rotated for 1 h at 4 °C. The cell lysates were subsequently gently rotated by a rotator overnight at 4 °C with 40 µL Ultra Link Immobilized NeutrAvidin protein (Pierce). Followed by the incubation, beads were washed 1× with lysis buffer LB3, 2× with LB2 (LB3 not composed of protease inhibitor), 2× with SWS containing 0.1% Triton X-100 in PBS (pH 7.4), 350 mM NaCl and 1 mM EDTA, and 1× with LB1 (LB2 not composed of 1% (*w*/*v*) Triton X-100). Then, the beads were completely mixed with 6 × Laemmli dye and boiled for 5 min before being loaded on SDS-PAGE gels.

### 4.14. Intracellular Ca^2+^ Measurement in Cell Populations

RAW-D cells (5 × 10^4^ cells) expressing GFP or GFP-Rab11b were seeded and grown in µ-dish 35 mm with an ibidi polymer coverslip bottom (Gräfelfing, Germany). After incubation with MEMα supplemented with RANKL (300 ng/mL) at 37 °C in 5% CO_2_ for 48 h, cells were successively washed with serum-free MEMα twice, and incubated for 60 min with the same medium composed of 1 μM Cal-590, a red-shifted Ca^2+^ indicator whose spectra of fluorescent emission is distinct from those of FITC, Alexa Flour 488, and GFP. Consequently, Cal-590 was ideally used to capture the distribution of Ca^2+^ signals in RAW-D cells expressing GFP with green fluorescence. The cells were then washed and analyzed using Olympus UPLSAPO 10X. The (Ca^2+^_i_) fluorescent intensities were measured and analyzed by Image J software.

### 4.15. Statistical Analysis

Unless otherwise noted, each sample was assayed in triplicate. For in vitro analyses, each experiment was thrice repeated. Statistical analysis was performed using Student’s *t*-test to compare two groups of independent samples (* *p* < 0.05, ** *p* < 0.01, and *** *p* < 0.001, n.s., nonsignificant). The level of statistical significance was set at 0.05 for all tests.

## Figures and Tables

**Figure 1 ijms-21-09352-f001:**
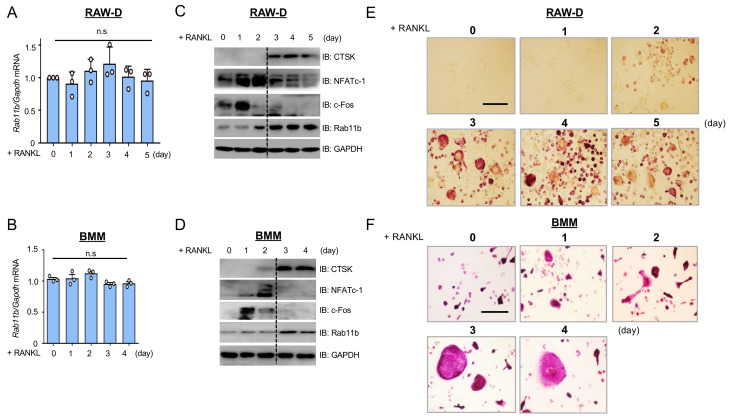
Rab11b upregulation at the late stage of osteoclastogenesis and the effect of RANKL (300 ng/mL) on osteoclast and monocyte/macrophage colony stimulating factor (M-CSF) (30 ng/mL) for a time course. (**A**,**B**) Total RNA was extracted, and cDNA was prepared from RAW-D cells or bone marrow-derived macrophages (BMMs) following a time course of RANKL (300 ng/mL) stimulation. Rab11b mRNA expression levels were analyzed by quantitative Polymerase Chain Reaction (qPCR). Mean ± SD of three independent repeats; n.s, nonsignificant (Student’s *t*-test), *n* = 3. (**C**,**D**) RAW-D cells or BMMs were treated with RANKL (300 ng/mL) over the indicated time course. Total expression levels of c-Fos, NFATc-1, CTSK, and Rab11b were assessed by immunoblotting and, GAPDH was used as a loading control. (**E**,**F**) TRAP-staining was carried out to assess osteoclast formation derived from RAW-D cells (**E**) and BMMs (**F**), following the indicated time course of RANKL stimulation. Bars 200 µm.

**Figure 2 ijms-21-09352-f002:**
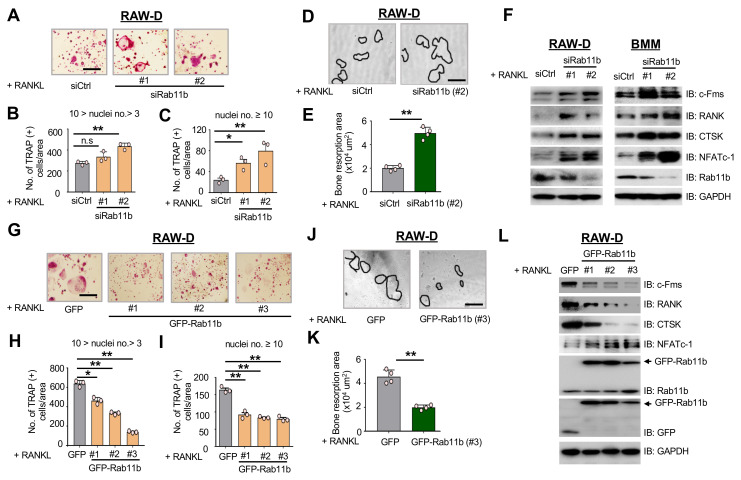
The effects of Rab11b knockdown and overexpression on osteoclastogenesis. (**A**) RAW-D cells were transfected with 10 pmol non-targeting siRNA (siCtrl) or two different types of Rab11b-specific siRNA (siRab11b #1 and #2) for 24 h. Then, the cells were cultured for an additional 3 days with media containing RANKL (300 ng/mL). TRAP-staining was performed to evaluate osteoclast formation. Bars 200 µm. (**B**) The number of TRAP-positive osteoclasts simultaneously having more than 3 nuclei and less than 10 nuclei per viewing field was counted. ** *p* < 0.01; n.s, nonsignificant (Student’s *t*-test); compared to the control. (**C**) The number of TRAP-positive osteoclasts having 10 or more nuclei per viewing field was counted. * *p* < 0.05; ** *p* < 0.01 (Student’s *t*-test); compared to the control. Mean ± SD of three independent repeats. (**D**) The images of the bone resorption area induced by RAW-D cell-derived osteoclasts transfected with siCtrl or siRab11b before seeded and cultured on the Osteo Assay StripWell upon RANKL (500 ng/mL) stimulation for 7 days. (**E**) The bone resorption area was measured and analyzed using Image J software. Mean ± SD of triplicate samples. ** *p* < 0.01 (Student’s *t*-test), *n* = 3. (**F**) After siRNA transfection for 24 h, RAW-D cells and BMMs were differentiated into osteoclasts by being cultured with RANKL (300 ng/mL) and a combination of RANKL (300 ng/mL) + M-CSF (30 ng/mL), respectively, for an additional 72 h. The total expression levels of c-Fms, RANK, NFATc-1, CTSK, and Rab11b were evaluated by immunoblotting, and GAPDH was used as a loading control. (**F**) The shown data were representative from two independent repeats. (**G**) RAW-D cells were transfected with retrovirus vectors either encoding Green Fluorescent Protein (GFP) as control or three different clones of GFP-Rab11b (abbreviated by #1, #2, and #3). The cultured cells were incubated with RANKL (300 ng/mL) for 3 days, and subsequently subjected to TRAP-staining analysis to assess the osteoclast formation. Bars 200 µm. (**H**) The number of TRAP-positive osteoclasts having simultaneously more than 3 and less than 10 nuclei was counted. Mean ± SD of three independent repeats. * *p* < 0.05, ** *p* < 0.01 (Student’s *t*-test); compared to the control. Bars 200 µm. (**I**) The number of TRAP-positive osteoclasts having 10 or more nuclei per viewing field was counted. Mean ± SD of three independent repeats. ** *p* < 0.01. (**J**) The bone-resorbing activities of the RAW-D-derived osteoclasts expressing GFP or GFP-Rab11b (clone #3). The cells were seeded and cultured on the Osteo Assay Stripwell Plates upon RANKL (500 ng/mL) stimulation for 10 days. (**K**) The images of the bone-resorption area were measured and analyzed using Image J software. Mean ± SD of the triplicate repeats. ** *p* < 0.01 (Student’s *t*-test) for the indicated comparisons. (**L**) The total expression levels of c-Fms, RANK, NFATc-1, CTSK, and Rab11b were evaluated by immunoblotting, and GAPDH was used as a loading control. The shown data were the representative from three independent repeats.

**Figure 3 ijms-21-09352-f003:**
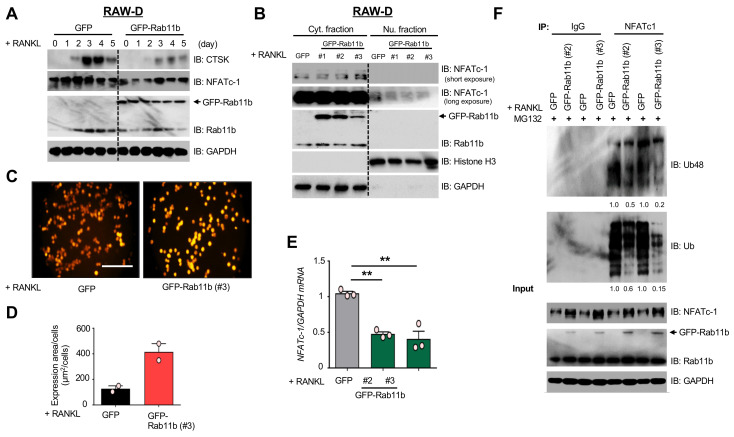
The effects of exogenous expression of Rab11b on NFATc-1 stabilization in response to (Ca^2+^_i_) elevation. (**A**) RAW-D cells expressing GFP or GFP-Rab11b (clone #3) was incubated with RANKL (300 ng/mL) over a time course, and subsequently harvested. The cell lysates were subjected to SDS-PAGE followed by immunoblotting for the detection of CTSK, NFATc-1, Rab11b, and GAPDH was used as a loading control. The representative data were obtained from two independent repeats. (**B**) Cytosolic and nuclear fractions were prepared from the RAW-D cell-derived osteoclasts expressing GFP or one of three different clones of GFP-Rab11b (#1, #2, and #3) upon RANKL (300 ng/mL) stimulation for 3 days. Cytosolic and nuclear fractions were subjected to SDS-PAGE and immunoblot analysis for detection of NFATc-1, Rab11b, and Histone H3 and GAPDH was used as the cytosolic and nuclear markers, respectively. The shown data were representative of two independent repeats. (**C**) (Ca^2+^_i_) oscillation in RAW-D cells expressing GFP or GFP-Rab11b (clone #3) in response to RANKL stimulation. After RANKL (300 ng/mL) addition for 2 days, the cells were washed by the serum (−/−) media, and subsequently loaded with 1 μM Cal-590 for 1 h. The cells were then washed and analyzed by Olympus Uplsapo 10X. *Bars* 100 µm. (**D**) The (Ca^2+^_i_) fluorescent intensities were measured and analyzed by Image J software. Mean ± SD of two independent repeats. (**E**) Quantitative Real Time-PCR (RT-PCR) analysis of NFATc-1 mRNA expression levels obtained from RAW-D cell-derived osteoclasts expressing GFP or one of two different clones of GFP-Rab11b (#2 and #3) after 3 days of RANKL (300 ng/mL) treatment. Mean ± SD of triplicate repeats. ** *p* < 0.01 (Student’s *t*-test), compared to the control (GFP). (**F**) Osteoclasts differentiated from RAW-D cell expressing GFP or one of two different clones of GFP-Rab11b (#2 and #3) upon RANKL (300 ng/mL) for 3 days. The cells were incubated with MG132 (10 μM for 4 h) before being lysed and subjected to anti-IgG or anti-NFATc-1 antibody IP followed by anti-Ub or anti-Ub48 with immunoblot analysis. The densitometer reading of polyubiquitinated NFATc-1 levels was measured underneath from 5th to 6th (for #2) or from 7th to 8th (for #3) lanes. The 5th or 7th lane was arbitrarily set as 1.0. The given results were obtained from two independent repeats.

**Figure 4 ijms-21-09352-f004:**
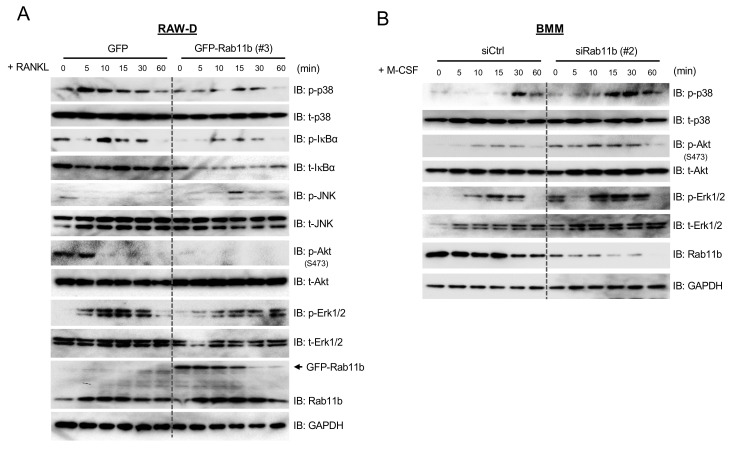
The effects of Rab11b on NFATc-1 upstream signaling cascades in macrophages stimulated by RANKL and M-CSF, respectively. (**A**) RAW-D cells expressing GFP or GFP-Rab11b (clone #3) were incubated with serum (−/−) culture media in the absence of RANKL. After RANKL (300 ng/mL) supplementation, the cells were incubated for the indicated times, and subsequently collected. The cell lysates were subjected to SDS-PAGE followed by immunoblotting for the detection of total expression levels of p-p38, t-p38, p-IκBα, t-IκBα, p-JNK, t-JNK, p-Akt (S473), t-Akt, p-Erk 1/2, t-Erk 1/2, Rab11b, and GAPDH was used as a loading control. The shown data were obtained from two independent repeats. (**B**) BMMs transfected with non-targeting siRNA (siCtrl) or Rab11b siRNA (siRab11b, type 2) were pre-incubated with serum (−/−) culture media in the absence of M-CSF. Then, the cells were incubated with M-CSF (50 ng/mL) for the indicated periods, and subsequently harvested. The cell lysates were subjected to SDS-PAGE followed by immunoblotting for the detection of p-p38, t-p38, p-Akt (S473), t-Akt, p-Erk 1/2, t-Erk 1/2, Rab11b, and GAPDH was used as a loading control.

**Figure 5 ijms-21-09352-f005:**
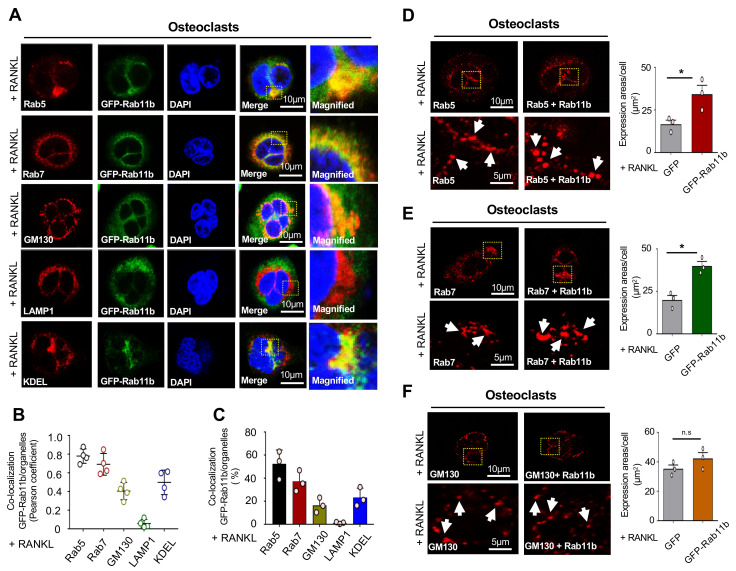
Subcellular localization of Rab11b in RAW-D cell-derived osteoclasts, and the effects of Rab11b overexpression on size-based modification of organelles. (**A**) The osteoclasts derived from RAW-D cells stably expressing GFP-Rab11b (clone #1) (green) were seeded on cover glasses with fixation and permeabilization of 0.2% Triton X-100 in PBS, and subsequently reacted with one of the antibodies against Rab5, Rab7, MG130, LAMP1, or KDEL (red, as indicated) that are specific markers for early endosomes, late endosomes, Golgi complex, lysosomes, or endoplasmic reticulum, respectively. Osteoclast DNA was stained with 4′,6-diamidino-2-phenylindole (DAPI) (blue). Specific regions of interest within a field are shown by yellow boxes in a magnified form on the left-hand top of each image it was taken from. Specific regions of interest within a field, designated by yellow boxes, are magnified on the right side of each image it was taken from. Mean ± SD of three independent repeats. Scale bar: 10 μm. (**B**,**C**) GFP-Rab11b co-localization with Rab5, Rab7, MG130, LAMP1 and KDEL was determined and evaluated by Pearson coefficient (**B**) or by color threshold analysis (**C**) using Fiji/ImageJ on at least 4 cells. (**D**–**F**) RAW-D cells stably expressing GFP or GFP-Rab11b were fixed and permeabilized with 0.2% Triton X-100, and stained with the specific antibodies to detect (**D**) Rab5 (red), (**E**) Rab7 (red), and (**F**) MG130 (red). The images were captured by confocal laser microscopy. The particle size was measured by (μm^2^) using ImageJ (shown on the right side). Mean ± SD of at least four independent repeats. * *p* < 0.05, n.s., nonsignificant (Student’s *t*-test).

**Figure 6 ijms-21-09352-f006:**
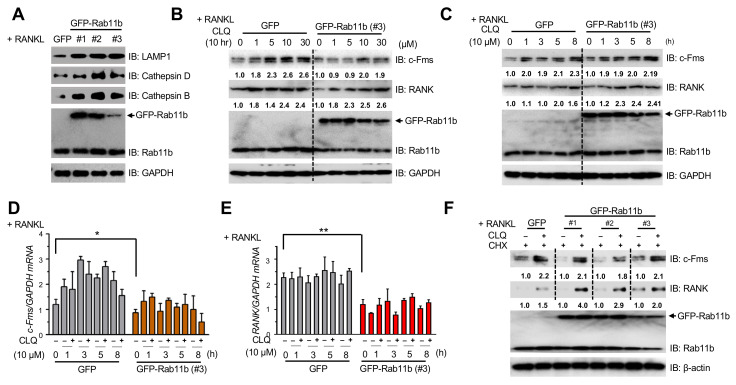
The effects of CLQ-mediated lysosomal function on endogenous levels of c-Fms and RANK in RAW-D cell-derived osteoclasts. (**A**) RAW-D cells expressing GFP or one of three different types of GFP-Rab11b (#1, #2, and #3) were treated with RANKL (300 ng/mL) for 3 days. The immunoblotting analysis of endogenous levels of LAMP1, a specific lysosomal receptor, and two lysosomal enzymes, Cathepsins B and D was done. GAPDH was used as a loading control. The shown data was representative of two independent repeats. (**B**) RAW-D cells expressing GFP or GFP-Rab11b (type #3) was pre-treated with RANKL (300 ng/mL) for 3 days before incubation with CLQ (0, 1, 5, 10, and 30 μM) for 10 h. The WB analysis of endogenous levels of c-Fms, RANK, and Rab11b was done. GAPDH was used as a loading control. The shown data was representative of two independent repeats. (**C**) RAW-D cells expressing GFP or GFP-Rab11b (type #3) were pre-treated with RANKL (300 ng/mL) for 3 days before incubation with CLQ (10 μM) over a time course (0, 1, 3, 5, and 8 h). The WB analysis of endogenous levels of c-Fms, RANK, and Rab11b was done. GAPDH was used as a loading control. (**D**,**E**) Total RNA was extracted, and cDNA was prepared from osteoclasts differentiated from RAW-D cells expressing GFP or GFP-Rab11b (type #3), following RANKL (300 ng/mL) stimulation for 3 days. Expression levels of c-Fms (**D**) and RANK (**E**) mRNAs were analyzed by qRT-PCR. Mean ± SD of two independent repeats. * *p* < 0.05, ** *p* < 0.01 (Student’s *t*-test). (**F**) RAW-D cells expressing GFP or one of three different types of GFP-Rab11b (#1, #2, and #3) were pre-treated with RANKL (300 ng/mL) for 3 days. The cells were subsequently treated with CHX (20 μg/mL) and simultaneously with or without CLQ (10 μM) for 5 h. The WB analysis of endogenous levels of c-Fms, RANK, and Rab11b was done. β-actin was used as a loading control. The shown data was the representative of two independent repeats.

**Figure 7 ijms-21-09352-f007:**
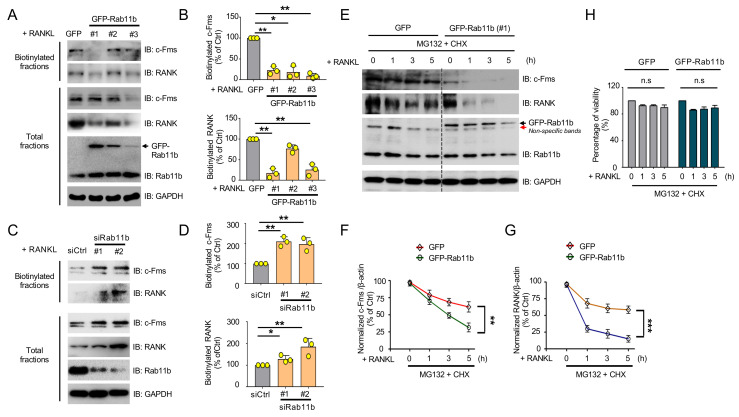
The effects of Rab11b overexpression on lysosome-mediated proteolysis of c-Fms and RANK receptors in osteoclasts. (**A**) RAW-D cells expressing GFP or three different types of GFP-Rab11b (#1, #2, and #3) were stimulated with RANKL (300 ng/mL) for 3 days. The cells were fractionated into two separate fractions including biotinylated fraction and total fraction as the protocol indicated in the “Material and Methods” section. The surface and total levels of c-Fms and RANK were evaluated by biotinylation pulldown and immunoblotting, respectively. (**B**) The densitometry quantification of bands indicating surface expression of c-Fms and RANK receptors was shown in the presence of GFP-Rab11b as a percentage of control (GFP). Mean ± SD of three independent repeats; * *p* < 0.05, ** *p* < 0.01 (Student’s *t*-test). (**C**) RAW-D cells were transfected with scrambled siRNA (siCtrl) or one of two different Rab11b siRNAs (Rab11b siRNA #1, Rab11b siRNA #2), followed by RANKL (300 ng/mL) stimulation for 3 days. The cells were fractionated into two separate fractions including biotinylated fraction and total fraction as the above protocol. The surface and total levels of c-Fms and RANK were evaluated by biotinylation pulldown and immunoblotting, respectively. (**D**) The densitometry quantification of bands indicating surface expression of c-Fms and RANK receptors was shown in the presence of Rab11b siRNAs as a percentage of control (siCtrl). Mean ± SD of three independent repeats; * *p* < 0.05, ** *p* < 0.01 (Student’s *t*-test). (**E**) RAW-D cells expressing GFP or GFP-Rab11b (#1) were stimulated with RANKL (300 ng/mL) for 3 days, followed by treatment with a combination of MG132 (20 μM) and CHX (20 μg/mL) for 0, 1, 3, and 5 h. Total levels of c-Fms and RANK were evaluated by immunoblotting. (**F**,**G**) The densitometry quantification of bands indicating total expression of c-Fms (**F**) and RANK (**G**) was shown as a percentage of the 1st lane for the GFP group referred to as the control (Ctrl), and the 5th lane for the GFP-Rab11b group referred to as the control (Ctrl). Mean ± SD of three independent repeats. ** *p* < 0.01, *** *p* < 0.001 (Student’s *t*-test). (**H**) Cell viability was assessed by the cellular ATP content measurement using the Cell Titer Glo Assay system. After stimulated with RANKL (300 ng/mL) for 3 days, RAW-D cells were co-treated with CHX (20 μg/mL) and CLQ (10 μM) over a time course. The values were the average of triplicate determinations with SD indicated by error bars. n.s., nonsignificant (Student’s *t*-test). The experiments were repeated thrice.

**Figure 8 ijms-21-09352-f008:**
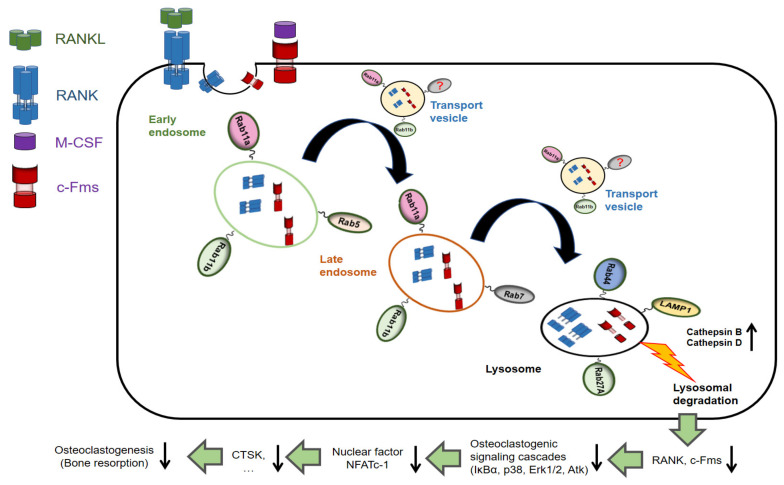
The model epitomizing the findings of our current works. c-Fms and RANK receptors were early internalized into early endosomes, and transported to late endosomes by the transport vesicles, which was subsequently fused with lysosomes for proteolysis of c-Fms and RANK surface receptors. Both homologs, Rab11a and b, upregulated at a late stage of osteoclast differentiation dictated the transport of c-Fms and RANK surface receptors to the lysosome via the axis of early and late endosomes-lysosomes. Rab11-mediated lysosomal proteolysis of c-Fms and RANK receptors sequentially weakened the osteoclastogenic signaling cascades, inhibited the nuclear translocation of the transcription factor, NFATc-1, reduced the expression level of CTSK, and eventually abolished osteoclastogenesis (bone resorption).

**Table 1 ijms-21-09352-t001:** List of sequences of DNA primers designed and used for quantitative real-time PCR.

Primer (Name)	Forward Sequence (5′→3′)	Reverse Sequence (5′→3′)
Rab11a	ACGTCATCTCAGGGCAGTTC	TTGGCTTGTTCTCAGTGGTG
Rab11b	AGAAGCTAAAAGCCCCTTGC	CAACTGGCCAGCGCGGAAAG
NFATc-1	TCATCCTGTCCAACACCAAA	TCACCCTGGTGTTCTTCCTC
c-Fms	TTGGACTGGCTAGGGACATC	GGTTCAGACCAAGCGAGAAG
RANK	CTTGGACACCTGGAATGAAGAAG	AGGGCCTTGCCTGCATC-3
GAPDH	ACCACAGTCCATGCCATCAC	TCCACCACCCTGTTGCTGTA

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
