# Peer review of "The Inhibitory Role of Rab11b in Osteoclastogenesis through Triggering Lysosome-Induced Degradation of c-Fms and RANK Surface Receptors"

_ijms, 2020, doi:10.3390/ijms21249352_

Round 1

Reviewer 1 Report

The manuscript numbered with title “The inhibitory role of Rab11b in osteoclastogenesis through triggering lysosome-induced degradation of c-Fms and RANK surface receptors” is an interesting study focused on critical role of Rab11b and its involvement in osteoclasts differentiation.

I recommend minor revisions for the manuscript. 

1. In the introduction section, the authors should better explain the aim and objectives of this work.

2. Figure 8 should be improved. It is confusing and it is difficult for the readers to understand the correct mechanism.

3. In the end of the discussion section the authors reported this sentence: “This study suggestive of how inhibitory regulators worked on osteoclast differentiation was beyond our knowledge of macrophagic features and osteoclastic bone resorption. Finally, our novel findings may contribute to facilitate the establishment of efficacious drugs and/or potential therapeutic methods for the treatment of macrophage-caused inflammatory diseases related to bone loss”. The potential application of these drugs or methods should be better clarified.

Author Response

We are truly appreciative of the reviewer’s constructive and insightful comments, according to which the manuscript has been carefully and rigorously revised. We hope the new version of our manuscript is now aptly suited for publication in International Journal of Molecular Sciences. The detailed responses to the Reviewer’ critiques and recommendations are shown below:  

  1. In the introduction section, the authors should better explain the aim and objectives of this work.

Response: We truly appreciate the reviewer’s great comments. The detail of our research objectives has been highlighted in the Introduction section at page 2 (from line 85 to line 93). Adding to this, we changed a little bit abstract section at page 1 (from lane 44 to lane 47).

  1. Figure 8 should be improved. It is confusing and it is difficult for the readers to understand the correct mechanism.

Response: We thank the reviewer for this vital comment, and totally agreed to the reviewer’s suggestion. We have analyzed the graphical abstract carefully and comprehensibly to clarify and simplify the Rab11-mediated transport pathway of c-Fms and RANK surface receptor via the axis of early endosomes-late endosomes-lysosomes, represented in Fig. 8. The descriptions of the graphical abstract were revised and given in the Fig. 8 legend.   

  1. In the end of the discussion section the authors reported this sentence: “This study suggestive of how inhibitory regulators worked on osteoclast differentiation was beyond our knowledge of macrophagic features and osteoclastic bone resorption. Finally, our novel findings may contribute to facilitate the establishment of efficacious drugs and/or potential therapeutic methods for the treatment of macrophage-caused inflammatory diseases related to bone loss”. The potential application of these drugs or methods should be better clarified.

Response: We thank reviewer’s crucial suggestions to which we totally agreed. Our prediction of future drug development program upon the probable treatment of bone-related diseases was given in the Discussion section at pages 15 (from line 376 to line 387). We strongly hope that our descriptions would meet the reviewer’s demands.     

Reviewer 2 Report

Manuscript IJMS 1026950

Reviewer comments

The manuscript entitled "The inhibitory role of Rab11b in osteoclastogenesis through triggering lysosome-induced degradation of c-Fms and RANK surface receptors" by Manh Tien Tran et al. presents original data on the impact of RAB11b expression levels onto the osteoclastogenesis and the osteoclast function in in vitro experiments. The presented results are really interesting and obtained with nicely designed experiments.

The Reviewer has no comment on the experimental part but an interrogation on the interpretation of the present data. More precisely the Reviewer wonders if the observed effects of RAB11b expression modulations on the osteoclastogenesis and the osteoclast function rather correspond to consequences of the global endosomes and lysosomes gain or lost of function than a direct effect on RANK and C-FMS expression at the cell surface. Say differently, the Reviewer believes that endosomes-lysosomes axis activity have to be limited during the osteoclastogenesis and high in mature osteoclasts to enable the resorption activity; As a major element of the endosomes-lysosomes functional machinery, RAD11b expression is crucial for the endosomes-lysosomes axis activity, and when RAD11b expression is perturbed this axis function is affected with consequences on either the osteoclastogenesis or the osteoclast activity according to respectively the increase or decrease expression levels. Can authors discuss this point and give arguments for a direct and specific effect of RAD11b on RANK and C-FMS steady state at the cell surface that was independent of the endosomes-lysosomes axis? For instance is the calcitonin receptor (or other receptors at the osteoclast cell-membrane) also affected by modulation of RAD11b expression or is it specific of C-FMS and RANK? The Reviewer will appreciate the answer to this interrogation to be included in the discussion of a revised version of the manuscript.

Author Response

We are truly appreciative of the reviewer’s constructive and insightful comments, according to which the manuscript has been carefully and rigorously revised. We hope the new version of our manuscript is now aptly suited for publication in International Journal of Molecular Sciences. The detailed responses to the Reviewer’ critiques and recommendations are shown below:  

The Reviewer has no comment on the experimental part but an interrogation on the interpretation of the present data. More precisely the Reviewer wonders if the observed effects of RAB11b expression modulations on the osteoclastogenesis and the osteoclast function rather correspond to consequences of the global endosomes and lysosomes gain or lost of function than a direct effect on RANK and C-FMS expression at the cell surface. Say differently, the Reviewer believes that endosomes-lysosomes axis activity have to be limited during the osteoclastogenesis and high in mature osteoclasts to enable the resorption activity; As a major element of the endosomes-lysosomes functional machinery, RAD11b expression is crucial for the endosomes-lysosomes axis activity, and when RAD11b expression is perturbed this axis function is affected with consequences on either the osteoclastogenesis or the osteoclast activity according to respectively the increase or decrease expression levels. Can authors discuss this point and give arguments for a direct and specific effect of RAD11b on RANK and C-FMS steady state at the cell surface that was independent of the endosomes-lysosomes axis? For instance is the calcitonin receptor (or other receptors at the osteoclast cell-membrane) also affected by modulation of RAD11b expression or is it specific of C-FMS and RANK? The Reviewer will appreciate the answer to this interrogation to be included in the discussion of a revised version of the manuscript.

Response: We strongly appreciate and thank reviewer’s crucial suggestions to which we totally agreed. It would be very important suggestions for our future works that we would like to further investigate the physiological role of Rab11b in regulating osteoclastogenesis. Because we observed the Rab11b localization in ER and Golgi apparatus in both RAW-D cells and osteoclasts, it was speculative of Rab11b to regulate other surface receptors of osteoclasts, required for osteoclastogenesis regulation. Therefore, we have provided the important details, suggested by the reviewer, at page 13 (from line 326 to line 335).